# VDR is an essential regulator of hair follicle regression through the progression of cell death

Yudai Joko[1,2], Yoko Yamamoto[3], Shigeaki Kato[4] (iD), Tatsuya Takemoto[5] (iD), Masahiro Abe[2], Toshio Matsumoto[1] (iD), Seiji Fukumoto[1], Shun Sawatsubashi[1,6,7] (iD)

**Vitamin D receptor (VDR) is essential for hair follicle homeostasis as its deficiency induces hair loss, although the mechanism involved remains unknown. Our research shows that, in *Vdr*-knockout mice, the hair cycle is halted during the catagen stage, preceding alopecia. In addition, in *Vdr*-knockout hair follicles, epithelial strands that normally regress during the catagen phase persist as "surviving epithelial strands." Single-cell RNA sequencing analysis suggests that these surviving epithelial strands are formed by cells in the lower part of the hair follicle. These findings emphasize the importance of the regression phase in hair follicle regeneration and establish VDR as a regulator of the catagen stage.**

## Introduction

Active vitamin D (VD) plays a key role in regulating calcium and phosphate metabolism in bones and the intestinal tract, and having other physiological effects through its interaction with the vitamin D receptor (VDR), a member of the nuclear receptor superfamily (Evans, 1988; Mangelsdorf et al, 1995; Li et al, 1997; Yoshizawa et al, 1997; Christakos et al, 2015). In the skin, which possesses the ability to produce VD, topical formulations of active VD are commonly used as a primary treatment for psoriasis, a chronic inflammatory skin disease (Mattozzi et al, 2016). Dysfunction of the VDR has also been linked to alopecia, as hair loss is a prominent feature in vitamin D-dependent rickets type 2A and mice lacking VDR (Chen et al, 2001; Bikle et al, 2006; Saini et al, 2017). However, mice lacking Cyp27b1, an active VD synthase, and thus lacking active VD, do not display alopecia (Hirota et al, 2017), suggesting that Vdr has an unknown function in hair follicle (HF) homeostasis that is independent of active VD. Despite this, the role of VD–VDR in the HF remains unclear and there is no established

treatment for alopecia in patients with vitamin D-dependent rickets type 2A. In addition, alopecia has not been reported in patients with point mutations inside the VDR ligand pocket, suggesting that a ligand-independent function of VDR regulates hair follicle homeostasis (Malloy & Feldman, 2011).

The HF, a mini-organ that emerges from epidermal basal cells, undergoes repeated cycles of growth (anagen), regression (catagen), and rest (telogen) throughout life (Müller-Röver et al, 2001; Alonso & Fuchs, 2006; Schneider et al, 2009; Lin et al, 2022). The HFs that initially form in the dorsal skin of the mouse develop on embryonic day 13.5, attain maturity by postnatal day (P)16, undergo the first catagen phase from P17, and enter the telogen phase at P20 (Fig S1) (Paus et al, 1999; Biggs & Mikkola, 2014). Single-cell RNA sequencing (scRNA-seq) analysis of HFs has shown that anagen follicles are composed of various cell types because of the expression of specific genes (Yang et al, 2017; Adam et al, 2018; Joost et al, 2020; Morita et al, 2021). However, scRNA-seq in catagen has yet to be reported. Advances in imaging technology have revealed that catagen progresses through a process of orderly cell death involving apoptosis and phagocytosis (Mesa et al, 2015). Wnt7b, Wnt10b, and Fgf18 have been identified as anagen-initiation factors and Fgf5 as an anagen-termination factor (Hébert et al, 1994; Kawano et al, 2005; Li et al, 2013; Kandyba & Kobielak, 2014); the regulator responsible for promoting cell death in catagen remains unknown. Moreover, no cases of impaired catagen progression in mouse models or patients with alopecia have yet been reported. In addition, previous reports have shown that VDR deficiency impairs β-catenin and Lef1-mediated Wnt signaling in keratinocytes, suggesting a role for VDR in anagen reentry (Cianferotti et al, 2007). Furthermore, VDR is expressed in the outer root sheath and dermal papilla (DP) of hair follicles, and its expression is known to increase from late anagen to catagen (Reichrath et al, 1994), but its function in catagen remains unknown. This study investigates the role of the VDR in the progression of catagen, and demonstrates that aberrant regression that ectopically prevents cell death impairs HF regeneration, emphasizing the significance of regression in tissue homeostasis.

[1]Department of Molecular Endocrinology, Fujii Memorial Institute of Medical Sciences, Institute of Advanced Medical Sciences, Tokushima University, Tokushima, Japan [2]Department of Hematology, Endocrinology and Metabolism, Institute of Biomedical Sciences, Tokushima University Graduate School, Tokushima, Japan [3]Department of Surgical Oncology, The University of Tokyo, Tokyo, Japan [4]Graduate School of Life Science and Technology, Iryo Sosei University, Fukushima, Japan [5]Laboratory for Embryology, Institute of Advanced Medical Sciences, Tokushima University, Tokushima, Japan [6]Research and Innovation Liaison Office, Institute of Advanced Medical Sciences, Tokushima University, Tokushima, Japan [7]Laboratory of Integrative Nuclear Dynamics, Institute of Advanced Medical Sciences, Tokushima University, Tokushima, Japan

Correspondence: shun-sawa2@umin.ac.jp

# Results and Discussion

### Deletion of VDR in the epidermis disrupts hair follicle homeostasis

To examine the function of VDR in the epidermis and HF separately from systemic calcium and phosphorus metabolism, we generated transgenic mice with conditional deficiency in epithelial *Vdr* (*Vdr* cKO) using *Keratin-14* (*Krt14*) promoter-driven Cre recombinase (Figs 1A and S2A and B). Epidermal deletion of Vdr resulted in progressive alopecia similar to that observed in *Vdr*-null mice by Xie et al (Fig 1A) (Yoshizawa et al, 1997; Xie et al, 2002). *Vdr* cKO mice with alopecia at 6 mo of age lost HFs and developed dermal cysts instead (Fig 1B). These dermal cysts did not express Krt6, which is known to mark the terminally differentiated companion layer and inner bulge, or Foxc1, which is expressed in the HF stem cells (HFSCs), inner root sheath (IRS), isthmus, and sebaceous gland of the HF (Figs 1C and S3A) (Hsu et al, 2011; Lay et al, 2016), but expressed Krt10 and Loricrin (Lor), markers of epidermal differentiation (Figs 1D and S3B), indicating that the dermal cysts are formed by epidermal-like cells. In addition, from around P60, *Vdr* cKO HFs lost dermal papilla (Fig S3C and D), gradually swelled, and transformed into dermal cysts (Fig S3E). To determine when the *Vdr* cKO mice lost their HFs, we examined skin tissues before the formation of dermal cysts. In *Vdr* cKO skin, normally morphogenic HFs were present at P15 (Fig 1E). However, *Vdr* cKO HFs did not enter anagen at P30 as compared with *Vdr* flox/flox (control) mice (Fig 1E). These results suggest that alopecia observed in *Vdr* cKO mice is caused by impaired entry into the first anagen phase and the transformation of the HF into a dermal cyst composed of epidermal cells (Fig 1E).

### Epidermal VDR regulates hair follicle regression

Despite normal morphogenesis during development, *Vdr* cKO mice failed to enter the first anagen, suggesting arrest of the hair cycle. To determine at what phase hair cycle arrest occurred in *Vdr* cKO mice, HFs were observed before P30. Both control and *Vdr* cKO HFs progressively shortened during the first catagen phase from P18 to P20 (Fig 2A). However, the position of the DP at P20 was above the adipocyte layer in control mice (Fig 2A, upper panel), but remained in the adipocyte layer in *Vdr* cKO mice (Fig 2A, lower panel). Two-photon microscopy images also showed that the lower part of control HFs was shortened and regressed at P19, whereas the lower part of the *Vdr* cKO HFs was elongated and had a folded structure (Fig 2B, Video 1, and Video 2). These findings indicated an abnormal regression phase in *Vdr* cKO mice after catagen V of the classical hair cycle stages (Müller-Röver et al, 2001). Therefore, to compare the HF regression after catagen V between control and *Vdr* cKO mice, HFs were classified into four categories (early, mid1, mid2, and late) based on the hair shaft depth (Fig 2C). It was predicted that there would be differences in the rate of cell death in the catagen phase because of the length of the lower part of the HF. To test this, cleaved caspase3 staining was performed. Three-dimensional microscopy images revealed that the average number of caspase3-positive cells was ~7–8 from mid1 to mid2 in control mice, whereas it was ~3–5 in the same period in *Vdr* cKO mice

(Figs 2C and D and S4A). In addition, epithelial strand length was increased in the *Vdr* cKO after mid2 as compared with control, whereas no significant alteration could be noticed in early and mid1 catagen (Fig S4B). As such, it is likely that, in *Vdr* cKO HFs, epithelial strands that should regress during the first catagen phase are retained as "surviving epithelial strands." These results suggested that Vdr promotes apoptosis in the lower part of the HF. Furthermore, as observed in *Vdr* cKO mice, surviving epithelial strands maintain a certain length even at the stage when control follicles enter telogen (Fig S4C and D), a state defined as "paused-catagen."

### Characteristic "surviving epithelial strands" in Vdr cKO mice are composed of Gata3+/Dst+/Krt14+ cells

To investigate the gene expression pattern of surviving epithelial strand formation in *Vdr* cKO mice, scRNA-seq was performed on epidermal and HF cells from the catagen phase (P18) of both control and *Vdr* cKO mice. The combined scRNA-seq data classified epidermal and HF cells into nine clusters (Fig 3A; Table S1). Approximately half of the total cells formed different clusters in control and *Vdr* cKO, suggesting that VDR plays a role in the identity of a wide range of epidermal and HF cells (Fig 3A and B). Based on the results of the density plot, clusters 1 and 2 were specific clusters for *Vdr* cKO mice. Cluster 1 was thought to be differentiated epidermal cells because of the high expression of *Lor*, and cluster 2 was then narrowed down as the candidate of surviving epithelial strand cells. Upon searching for characteristic gene markers of cluster 2, *Dst* and *Gata3* were identified as potential candidates (Fig 3C). Immunofluorescence staining (IF) confirmed that surviving epithelial strands could be marked with Dst and Gata3 (Fig 3D). Although Gata3 has previously been used as a marker for IRS in anagen HFs (Kaufman et al, 2003), IRS markers other than Gata3 did not mark surviving epithelial strands (Fig S5A and B). In addition, IF showed that Gata3 was strongly expressed in IRS and weakly expressed in basal cells (Krt14-positive) in the lower part of HFs (lower proximal cup: LPC) from late anagen to early catagen (Inês & Nicolas, 2012). During catagen progression, cells with low Gata3 expression formed surviving epithelial strands, whereas IRS moved upward (Fig 3E and F). However, within the *Vdr* cKO HF, cells expressing low levels of Gata3 exhibited resistance to apoptosis, specifically in the epithelial strands (Fig S5D). VDR and Dst, which are known to be specifically expressed in LPC of anagen (Yang et al, 2017), were also expressed in the epithelial strand of control mice (Figs 3D and S5C). These results suggested that VDR is a crucial factor for the elimination of the LPC cells expressing Dst, Gata3, and Krt14 during catagen. However, in the *Vdr* cKO HF, some Gata3+/Dst+/Krt14+ LPC cells are eliminated, and certain cells that escape cell death form surviving epithelial strands (Fig 3F). Moreover, Dst function in HF is unknown.

### Hair-plucking stimulation bypasses the paused-catagen state and allows entry into the anagen phase

*Vdr* cKO HFs exhibited a failure to eliminate the epithelial strand, leading to an inability to transition into the anagen phase and a persistent state in paused catagen (Fig 1E). Nevertheless, immunofluorescence assays revealed the presence of HFSCs identified as

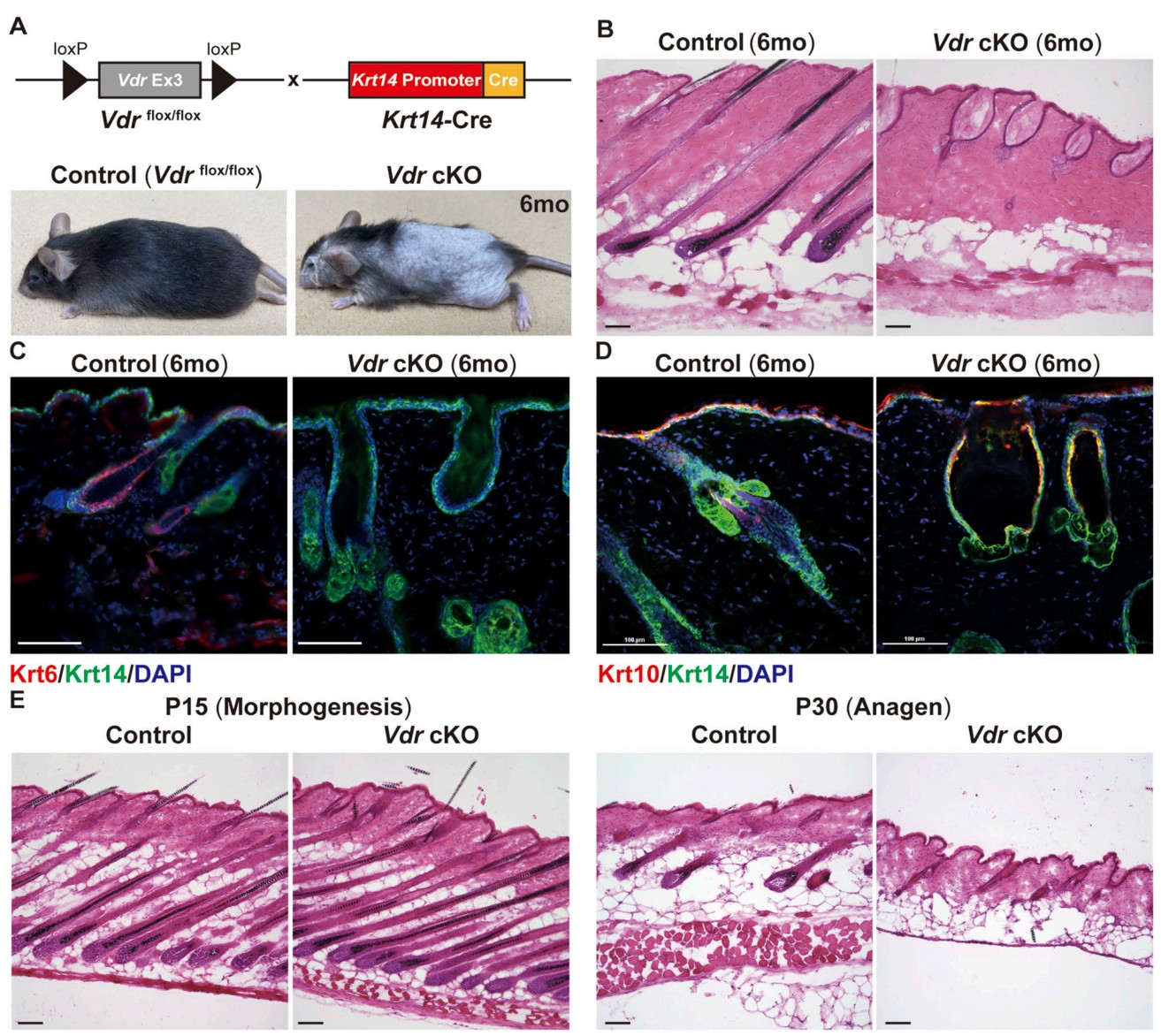

**Figure 1. Deletion of Vitamin D receptor (VDR) in the epidermis disrupts hair follicle homeostasis.**
**(A)** Schematic representation of engineered alleles to establish epidermis-specific *Vdr*-deficient mice. Images of 6-mo-old *Vdr* flox/flox and *Vdr* cKO mice.
**(B)** Hematoxylin and eosin (HE) staining of 6-mo-old mouse skin sections. Scale bar, 100 µm. **(C)** Immunofluorescent (IF) staining for the inner root sheath layer marker Krt6 (red) and the basal layer marker Krt14 (green). Scale bar, 100 µm. **(D)** IF staining for the differentiated suprabasal layer markers Krt10 (red) and Krt14 (green). Scale bar, 100 µm. **(E)** HE staining of morphogenesis-stage (postnatal day (P)15) and anagen-stage (P30) mouse skin sections. Scale bar, 100 µm.

Sox9-positive and the bulge region as Col17-positive within the HFs, even in this paused-catagen state (Fig 4A and B). In other words, it is suggested that HFSCs are alive even in the paused-catagen state. Based on this observation, we hypothesized that the presence of surviving epithelial strands in *Vdr* cKO mice impeded reentry into the anagen phase, despite the preservation of normal HFSCs. Hence, we explored the possibility of inducing the anagen phase in *Vdr* cKO HFs by triggering HFSC activation. We performed hair plucking (HP) at P20, which stimulates HFSC activation via macrophage-derived TNFα (Rahmani et al, 2020), and observed that anagen reentry was successfully triggered in *Vdr* cKO HFs 7 d after HP (Figs 4C and S6A and B). *Wnt10a, Wnt10b* (expressed in HFSCs),

*Fgf7* and *Fgf10* (expressed in DP) are known to be activated during the transition to anagen phase, so we examined whether their expression levels were altered by HP. In control mouse skin, *Wnts* and *Fgfs* expression levels were low at P22 (telogen phase), but increased at P27 (anagen phase). In contrast, in *Vdr* cKO mouse skin, *Wnts* and *Fgfs* expression levels were low at both P22 (paused-catagen) and P27 (paused-catagen), but HP increased these expression levels to the same levels as in the anagen phase of control mice (Fig 4D). In addition, using Wnt reporter mice (Takemoto et al, 2016), it was found that HP induced Wnt activation in *Vdr* cKO HFs (Fig 4E). These findings indicate that, once in a paused-catagen state, *Vdr* cKO mice are unable to initiate the subsequent hair cycle

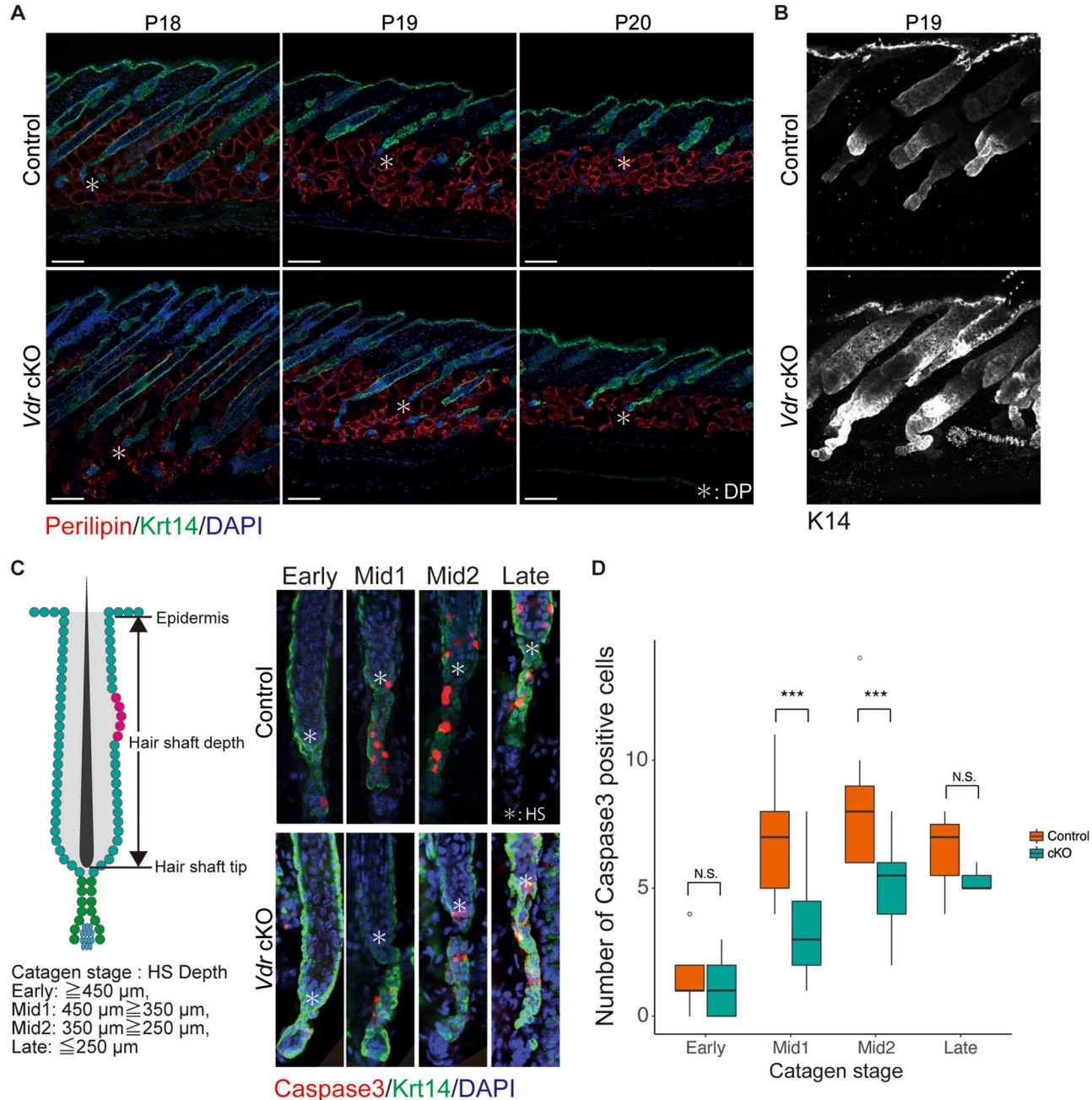

**Figure 2. Epidermal Vitamin D receptor regulates hair follicle regression.**
**(A)** IF staining of P18–P20 mouse skin sections for the adipocyte markers perilipin (red) and Krt14 (green). Scale bar, 100 $\mu m$. *, dermal papilla (DP). **(B)** IF staining of P19 mouse skin section for Krt14 (white). **(C)** IF staining of early- to late-catagen hair follicle of back skin for the apoptosis markers caspase3 (red) and Krt14 (green). Catagen stage = hair shaft (HS) depth; early: ≥450 $\mu m$, mid1: 450 ≥ 350 $\mu m$, mid2: 350 ≥ 250 $\mu m$, late: ≤250 $\mu m$. **(D)** Number of caspase3-positive cells per hair follicle (n = 3) at each catagen stage. Error bars, mean ± SEM; ***$P < 0.001$ ($t$ test).

without external intervention. However, these mice retain the re-generative potential of their HFSCs and DP even in the paused-catagen state. The results further suggest that external activation of the *Wnts*- and *Fgfs*-signaling pathways can rescue *Vdr* cKO HFs from the paused-catagen state before dermal cyst formation and indicate that the formation of surviving epithelial strands may inhibit the activation of HFSCs.

**VDR plays a key role in the process of hair follicle regression**

This study sheds light on the role of the VDR in homeostasis of the HF, which is deemed to be a regenerative mini-organ of the skin. In *Vdr* cKO mice, we found that the hair cycle was arrested in the middle of the catagen phase before the onset of alopecia (Fig 2). In addition, surviving epithelial strands in paused catagen are of a

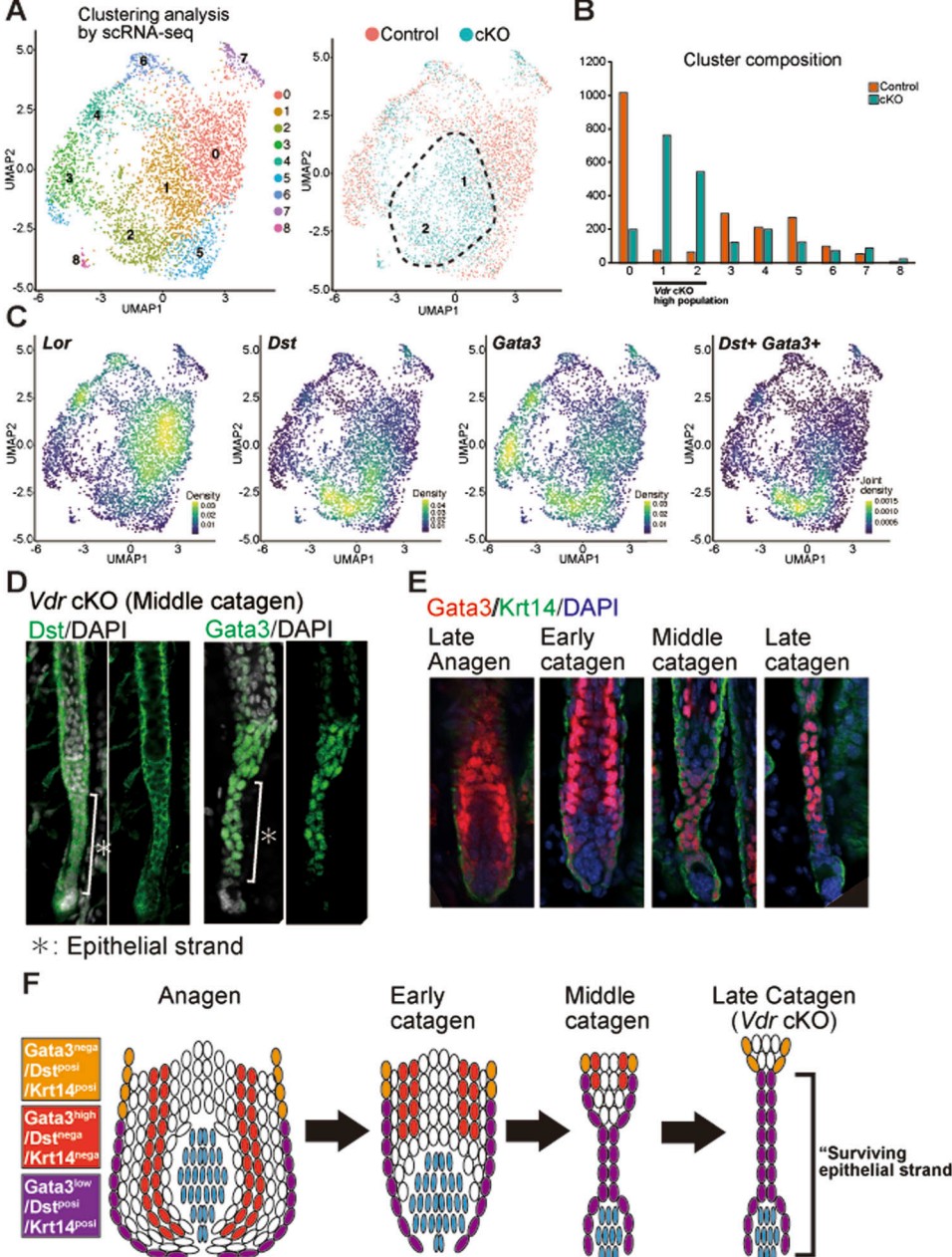

**Figure 3. The characteristic "surviving epithelial strands" in Vitamin D receptor cKO mice are composed of Gata3+/Dst+/Krt14+ cells.**
**(A)** Left: P18 control (n = 2,097) and cKO (n = 2,140) epidermal cell transcriptomes visualized with UMAP plot, colored according to unsupervised clustering. Right: UMAP plot colored red (control) or blue (cKO). See Table S1 for cluster markers list. **(B)** Number of cells in each cluster. **(C)** Density plots showing the RNA expression of Lor, Dst, Gata3, and Dst&Gata3. **(D)** IF staining of middle-catagen hair follicle for the cluster2 marker Dst (Left) and Gata3 (right). **(E)** IF staining of late-anagen to late-catagen hair follicle for Gata3 (red) and Krt14 (green). **(F)** Scheme of the "surviving epithelial strand" formation by Gata3-low/Dst+ cells.

certain length in all HFs, suggesting that they are formed by a specific cell population (Fig S4C and D). Our findings suggest that proper elimination of disused cells is a crucial aspect of this regenerative process for tissue homeostasis. During the catagen phase, HFs typically undergo regression through programmed cell death, bringing the bulge and DP closer together and activating *Wnt* and *Fgf* signaling, which initiates the growth phase (Fig 5 left panel) (Rendl et al, 2008; Oshimori & Fuchs, 2012). However, in *Vdr* cKO mice, the rate of cell death during the catagen phase is reduced, leading to the formation of "surviving epithelial strands" and preventing the completion of the catagen phase. Consequently, HFs deficient in *Vdr* become entrapped in a "paused-catagen" state,

with *Wnt* and *Fgf* signaling being inactivated, eventually transforming into dermal cysts and resulting in the irreversible disruption of tissue homeostasis (Fig 5 upper right). Meanwhile, *Vdr* cKO HFSCs retain their regenerative capacity until the inception of dermal cyst formation, and can be activated by HP stimuli (Fig 5 lower right). These results provide a novel paradigm for alopecia because of catagen arrest and emphasize the importance of the regression phase in reactivating HFSCs and regenerating HFs. In addition, it has been reported that most patients with vitamin D-dependent rickets type 2A are born with a normal hair distribution followed by hair loss between 1 and 3 mo of age, suggesting that *VDR*-deficient alopecia in humans may also be attributable to

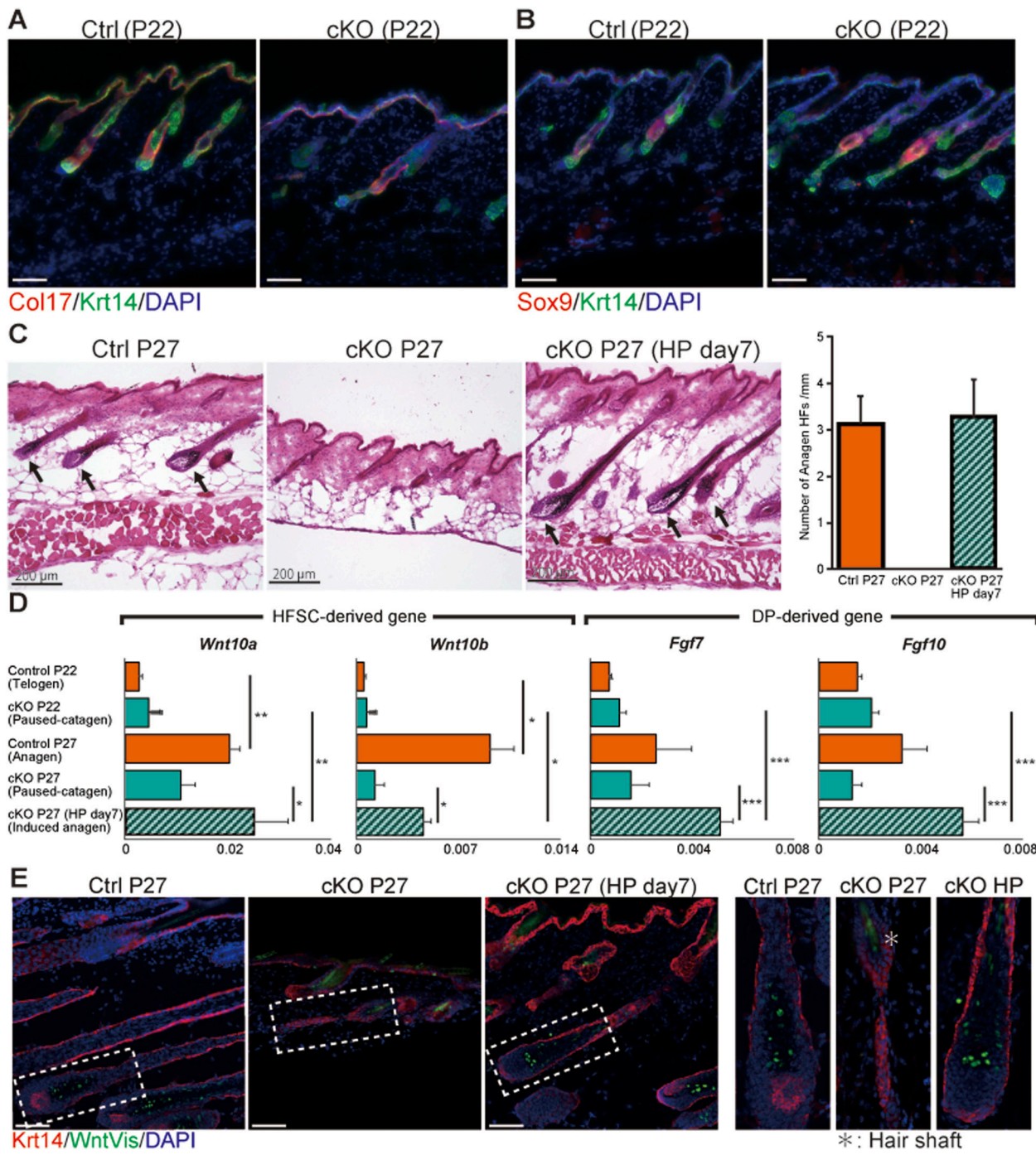

**Figure 4. Hair-plucking stimulation bypassed the paused-catagen state and allowed entry into the anagen phase.**
**(A)** IF staining of P22 mouse skin section for the bulge markers Col17 (red) and Krt14 (green). **(B)** IF staining of P22 mouse skin section for the hair follicle stem cell markers Sox9 (red) and Krt14 (green). **(C)** HE staining of mouse skin sections (control, cKO, and 7 d after hair plucking). Scale bar, 200 μm. Number of anagen hair follicles per millimeter in each mouse. **(D)** RT–qPCR analysis of *Wnt10a*, *Wnt10b*, *Fgf7*, and *Fgf10* mRNA expression in control (P22) (n = 3), cKO (P22) (n = 3), control (P27) (n = 3), cKO (P27) (n = 3), and cKO (7 d after hair plucking) (n = 3) mouse skin. **(E)** Wnt reporter expression (green) in mouse skin section. *autofluorescence of hair shaft. Data information: in (D), data are presented as mean ± SEM; *P < 0.05; **P < 0.01; ***P < 0.001 (Welch's *t* test).

catagen arrest. Furthermore, it may be feasible to treat alopecia in patients with vitamin D-dependent rickets type 2A by either inducing activation of the signaling between HFSCs and DP by external stimuli or by eliminating surviving epithelial strands before cyst formation.

We found that the inception of hair cycle arrest because of *Vdr* deficiency involves the formation of surviving epithelial strands. Previous studies using *Vdr*-deficient mice and their supposed phenocopy Hr/Hr mice suggested that the connection between the

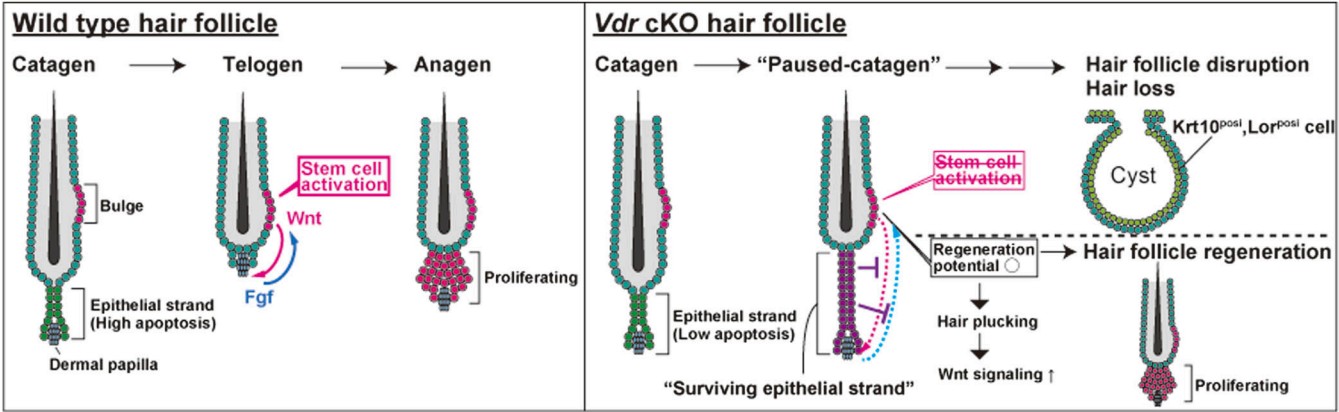

**Figure 5. Schematic of catagen to anagen transition in WT or *Vdr* cKO.**
At the catagen of WT hair follicles, epithelial strand cells are actively eliminated by apoptosis. When an epithelial strand is eliminated, interactions between HF stem cells (HFSCs) and DP, such as Wnt and Fgf signaling, activate HFSCs and transit to anagen. Meanwhile, *Vdr*-cKO hair follicles have a low rate of apoptosis in epithelial strands, resulting in the formation of a "surviving epithelial strand." Inhibition of HFSC activation by this "surviving epithelial strand" results in arrest of the hair cycle, loss of hair follicles, and cyst formation. However, *Vdr*-cKO mice retain their regenerative potential even in the "paused-catagen" state. Therefore, when HFSC activation is promoted by HP stimulation, hair follicles regenerate.

HF and the DP is lost around P22 (Mann, 1971; Panteleyev et al, 1999; Bikle et al, 2006). However, our approach of tissue clearing and three-dimensional imaging showed that surviving epithelial strands maintain the connection between HFs and DP (Fig 2B, Video 2). It was previously asserted that VDR deficiency-induced alopecia was correlated with the reduction of HFSCs in mice older than 3.5 mo (Cianferotti et al, 2007), yet our findings suggest that the formation of surviving epithelial strands at earlier time points may contribute to alopecia. By applying tissue clearing and three-dimensional imaging, we have succeeded in detecting previously unknown morphological abnormalities, and this method is promising for detecting tissue disruption, leading to early hair loss. Our scRNA-seq in the catagen phase (Fig 3), a novel approach, uncovered that surviving epithelial strands comprise of Gata3+/Dst+/Krt14+ cells. We have observed that this population of cells is present in the lower part of late-anagen HFs, suggesting that LPC-derived cells may form surviving epithelial strands. Future studies by live imaging using reporter mice are needed to determine the origin of cells forming surviving epithelial strands.

Our findings highlight the importance of regression in HF regeneration and demonstrate that the VDR is a novel regulator of the catagen phase. Although Fgf5 has been reported as a factor that regulates catagen (Hébert et al, 1994), it is specifically a factor that controls entry into catagen, whereas VDR is a novel factor that controls catagen progression. It has recently become clear that catagen progression is coordinated by the contraction of the dermal sheath smooth muscle (Heitman et al, 2020; Martino et al, 2023) and orderly cell death involving apoptosis and phagocytosis (Mesa et al, 2015). We confirmed that the VDR is expressed in the nuclei of normal epithelial strand-forming cells during the catagen phase, suggesting that the VDR may promote the cell death of epithelial strands through transcriptional regulation. Accumulation of apoptotic and senescent cells has recently been shown to contribute to the development of inflammatory and age-related diseases, highlighting the significance of proper cell elimination for maintaining tissue homeostasis (Poon

et al, 2014; Scudellari, 2017; Chaib et al, 2022). Our findings suggest that cell elimination by the VDR during catagen also plays a crucial role in HF homeostasis. Future studies should clarify the molecular function of VDR to promote cell death in the catagen. Meanwhile, Morita et al reported that *Vdr* was strongly expressed in the lower part of the HF at E13.0–E17.0 when these cells were produced (Morita et al, 2021). It is thus possible that the cell fate during catagen is determined via VDR at the stage of HF development. Either way, the molecular function of VDR in HF homeostasis requires further investigation.

# Materials and Methods

### Mice

Mice with *Vdr* knockout specifically in epidermal cells were generated by breeding *Vdr* flox/flox mice (Yamamoto et al, 2013) and Krt14cre mice (Dassule et al, 2000). Wnt reporter mice were as previously described (Takemoto et al, 2016). The care and handling of animals in this study were in accordance with the guidelines set forth by Tokushima University for animal and recombinant DNA experiments. Offspring were genotyped via PCR-based assays of mouse-tail DNA (Fig 2A and B). Primer sequences are listed in Table S2.

### Frozen sections

For two-dimensional imaging, dorsal skin tissues were embedded in an optimal cutting temperature (OCT) compound (Sakura Finetechnical) and stored at –80°C. The frozen samples were then cut into 15-µm-thick sections using a cryostat (CM1860; Leica).

For three-dimensional imaging, dorsal skin samples were fixed with 4% PFA in PBS for 10 min at RT. The fixed tissues were washed with PBS, embedded in Othe CT compound, and stored at –80°C. The frozen samples were then cut into 100-µm-thick sections using a cryostat.

### Hematoxylin and eosin (HE) staining

Frozen sections were removed from the OCT compound and fixed in 4% PFA in PBS, and then stained with hematoxylin (131-09665; FUJIFILM Wako Pure Chemical) and eosin for analysis of tissue histology using an all-in-one fluorescence microscope (BZ-X700; KEYENCE).

### Immunofluorescence (IF) staining for two-dimensional imaging

For two-dimensional imaging, frozen sections were removed from the OCT compound, fixed in 4% PFA in PBS, and then blocked with a blocking buffer (5% normal donkey serum, Blocking One [03953-95; Nacalai Tesque, Inc.]) for 1 h at RT, followed by incubation with a primary antibody overnight at 4°C. The following day, the sections were washed in 0.2% Triton X-100 in PBS and incubated with the corresponding secondary antibodies. The sections were then washed in 0.2% Triton X-100 in PBS and mounted. All antibodies and dilutions are listed in Table S3. Sections were counterstained with DAPI (0100-20; SouthernBiotech) to visualize nuclei. All fluorescence microscopy images were captured using an all-in-one fluorescence microscope or confocal microscope (A1R; Nikon).

### Immunofluorescence (IF) staining for three-dimensional imaging

For three-dimensional imaging, 100-$\mu$m sections were permeabilized with 0.1% Digitonin (300410; Merck) in PBS for 30 min, blocked with the blocking buffer (5% normal donkey serum, Blocking One) for 1 h at RT, and then incubated with a primary antibody overnight at 4°C. The following day, sections were washed in 0.2% Triton X-100 in PBS and incubated with the corresponding secondary antibodies. The sections were then washed in 0.2% Triton X-100 in PBS and mounted. All antibodies and dilutions are presented in Table S3. Sections were counterstained with Hoechst (H3570; Thermo Fisher Scientific) to visualize nuclei. All fluorescence microscopy images were captured using a confocal microscope (A1R) or multi-photon microscope (A1R MP; Nikon).

For multi-photon microscope imaging, stained 100-$\mu$m sections were transferred to a custom-made clearing solution A (20% thiodiethanol, 24% sucrose in ultrapure water) for 1 h at RT and then mounted in a custom-made clearing solution B (45% thiodiethanol, 5% glycerol, and 50% iomeprol [877219; Eisai Co., Ltd.]). These custom-made reagents were based on the LUCID clearing agent protocol (Sawada et al, 2018), with slight modifications. Cleared tissue samples were then imaged using a multi-photon microscope (A1R MP).

### RNA preparation and quantitative RT–PCR

Total RNA was isolated from 5 mm$^2$ of mouse dorsal skin using RNAiso Plus (9109; Takara), and cDNA was synthesized using ReverTra Ace qPCR RT Master Mix (FSQ-301; Toyobo), in accordance with the manufacturer's protocol. Quantitative PCR analysis was performed using a LightCycler 96 (Roche) with FastStart Essential DNA Green Master (Roche). Results were calculated as mean ± SD from at least three independent experiments. Primer sequences are listed in Table S2.

### Isolation of epidermal and hair follicle cells and single-cell library preparation

To isolate epidermal and hair follicle cells, P18 dorsal skins from *Vdr*-cKO and control mice were collected and placed dermis down in 0.25% trypsin (25200072; Thermo Fisher Scientific) for 20 min at 37°C. Cell suspensions were obtained by gently scraping the skin. The cells were then filtered with strainers (70 and 40 $\mu$m) (542070 and 542040; Greiner Bio-One). The cell viability ratio was confirmed to be 80% or more by trypan blue staining.

Dorsal skin samples of three male mice were prepared for constructing one library. RNAdia 2.0 kit (Dolomite Bio) was used for scRNA-seq library preparation following the manufacturer's protocol. Briefly, cells were loaded into a Nadia (Dolomite Bio) microfluidics cartridge at a concentration of 300 cells per microliter. Cells were lysed in a droplet, and emulsion formed from the microfluidics device was then isolated and droplets were broken with 1H,1H,2H,2H-perfluoro-1-octanol. Reverse transcription was then performed, and purified cDNA was used as an input for Nextera tagmentation reactions.

### Single-cell RNA sequencing analysis

Single-cell RNA-seq analysis was performed on NovaSeq using an S4 flow cell with a PE 150 kit (Illumina). Sequencing datasets were aligned using Seurat v4 (Hao et al, 2021). The Seurat v4 guidelines were followed for the identification of variable genes, dimensionality reduction, and cell clustering. A resolution of 0.5 was used for parameter identification. The Wilcoxon statistical test built into Seurat v4 was used to identify markers.

### Hair plucking

The dorsal hair of P20 mice was plucked using tweezers to induce anagen.

## Data Availability

The scRNA-seq data have been deposited in GEO under accession code GSE223884.

## Supplementary Information

## Acknowledgements

We would like to express our gratitude to N Miura, R Sakai, A Teraoku, and N Itoyama for technical assistance at Fujii Memorial Institute of Medical Sciences. We also thank all the members of Fujii Memorial Institute of Medical Sciences for their support and use of the facilities in this research. We are also grateful to Edanz (https://jp.edanz.com/ac) for editing a draft of this article. This work was supported in part by a grant from the Setsuro Fujii

Memorial, the Osaka Foundation for the Promotion of Fundamental Medical Research, and the Ministry of Education, Culture, Sports, Science and Technology and Japan Society for the Promotion of Science (KAKENHI) Grant No. 21J12961, 16K19556, and 18K19518.

## Author Contributions

Y Joko: data curation, formal analysis, funding acquisition, investigation, methodology, and writing—original draft.
Y Yamamoto: resources.
S Kato: resources.
T Takemoto: resources.
M Abe: supervision and investigation.
T Matsumoto: supervision, investigation, and writing—review and editing.
S Fukumoto: supervision, investigation, and writing—review and editing.
S Sawatsubashi: conceptualization, data curation, formal analysis, funding acquisition, investigation, methodology, project administration, and writing—original draft, review, and editing.

## Conflict of Interest Statement

The authors declare that they have no conflict of interest.

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
