## [Reviewer comments · Life Science Alliance]

Life Science Alliance

VDR is an essential regulator of hair follicle regression through the progression of cell death

Yudai Joko, Yoko Yamamoto, Shigeaki Kato, Tatsuya Takemoto, Masahiro Abe, Toshio Matsumoto, Seiji Fukumoto and Shun Sawatsubashi

DOI: <https://doi.org/10.26508/lsa.202302014>

Corresponding author(s): Dr. Shun Sawatsubashi (Institute of Advanced Medical sciences, Tokushima University)

Review Timeline:

Submission Date:	2023-02-27
Editorial Decision:	2023-04-06
Revision Received:	2023-07-03
Editorial Decision:	2023-08-10
Revision Received:	2023-08-14
Accepted:	2023-08-14

Transaction Report:

April 6, 2023

Re: Life Science Alliance manuscript #LSA-2023-02014-T

Shun Sawatsubashi
Tokushima University, Institute of Advanced Medical Sciences

Dear Dr. Sawatsubashi,

Thank you for submitting your manuscript entitled "VDR is an essential regulator of hair follicle regression through the progression of cell death" to Life Science Alliance. The manuscript was assessed by expert reviewers, whose comments are appended to this letter. We invite you to submit a revised manuscript addressing the Reviewer comments.

Thank you for this interesting contribution to Life Science Alliance. We are looking forward to receiving your revised manuscript.

Sincerely,

B. MANUSCRIPT ORGANIZATION AND FORMATTING:

Reviewer #1 (Comments to the Authors (Required)):

The authors explain that despite alopecia is commonly observed in VDDR2A patients and in mice lacking VDR, there is another mouse model lacking *Cyp27b1* (and thus lacking active vitamin D) that does not display alopecia. From this fact stems the elegant hypothesis that an additional unknown function of VDR in hair follicle (HF) dynamics exists, possibly related to the modulation of catagen progression.

First, using K14 Cre Vdr cKO mice the authors narrow down the onset of HF morphological alterations to a timepoint after P15 and preceding the first postnatal anagen stage at P30. Further morphological analyses of tissue sections determined that the abnormalities follow the catagen V phase of the HF cycle. Differential length of the terminal end of the HFs in catagen motivated the study of cleaved caspase 3 expression, identifying a decrease in the number of positive cells for this marker in Vdr cKO mice coincident with retained longer epithelial strands. The scRNAseq analysis of the dorsal epidermis suggested that Vdr is involved in the elimination of the epithelial strands expressing *Gata3+/Dst+/Krt14+*, although this process is not completely impaired in Vdr cKO mice, in which only some cells of this type escape cell death and remain forming the surviving epithelial strands. The manuscript includes no validation of this cell population undergoing a lower level of cell death in cKO mice. Plucking-induced anagen entry occurred normally in Vdr cKO mice, indicating that the failure in hair regeneration can be bypassed.

The manuscript is well-written and structured.

Major points:

- The results of this study support that the impairment in catagen progression is related to the Vdr deficiency. However, the authors suggest that this effect is independent of the interaction of the ligand (active vitamin D) with its receptor. Even though the manuscript is mostly descriptive, can the authors provide data demonstrating that the defect in HF regeneration derived from the impaired progression of the catagen phase is not reproduced by blocking the interaction of active vitamin D and its receptor in the skin of control mice?
- Fig. 3 showing the "surviving epithelial strands" during the catagen stage should combine the identity markers used to define the cell subpopulation of interest at least with the cleaved caspase 3 marker. Additional approaches in this regard such as flow cytometry analysis could help to reinforce this point.
- Indicate clearly the number of animals included in the experimental groups for each experiment. This is needed to evaluate the strength of the data presented. The statistical analyses that have been used should also be clearly stated.

Minor points:

- Indicate in the text which specific Wnt and Fgf are studied at the expression level as it is detailed in Fig. 4.

Note to the Editor: I have not been able to find the supplementary figures.

Reviewer #2 (Comments to the Authors (Required)):

In this manuscript, Yudai Yoko et al., suggest the way VDR regulates the hair follicle regression. The manuscript nicely written and clearly depicts the role of VDR in regulating the hair follicle at catagen stage by showing the presence of less cell death, leaving behind the extended epithelial strand which the author calls a "surviving epithelial strand". They further elucidate with the help of scRNA analysis how VDR helps in eliminating the cells expressing *Dst*, *Gata3* and *K14* in the in the lower proximal cup region. Finally to confirm the role of VDR and the said genes in regulating the hair cycle continuation they conduct a hair plucking experiment in both control and the VDR cKO mice before the catagen stage to show indeed the process takes place due to reactivation of Fgf and Wnt under the influence of VDR.

In my opinion the manuscript is written well with subsequent results to prove the role of VDR in maintaining catagen stage hair follicle (HF). I strongly favor to accept the manuscript with said few minor changes in text or after minor clarifications.

Please change the below mentioned text or update with relevant answers:-

- "The HFs that initially form in the dorsal skin of the mouse develop on embryonic day 12"---- I find this statement to be wrong as the mouse hair placode starts developing at E13.5-Please correct with right publication reference (Eg:- Biggs & Mikkola 2012)
- "These dermal cysts did not express *Krt6* and *Foxc1*, which are expressed in the inner root sheath (IRS) of the HF (Fig. 1C, Supplemental Fig. S3A), but expressed *Krt10* and *Loricrin (Lor)*, markers of epidermal differentiation (Fig. 1D, Supplemental Fig.

S3B), indicating that the dermal cysts are formed by pidermal cells" -----This context do not explain properly. Does it mean the Krt6 and Foxc1 are not expressed by epidermis? Foxc1 is known to be a HFSC marker (expressed in ORS).

- What kind of connection is between Wnt, Fgf and Vdr? Is it directly connected to something else which is triggering the activation of Fgf and Wnt during Anagen phase?
- I would be happy if the author also check the expression of NFATc1 by immunostain in VDR cKO expression during different stages of hair cycle.

Reviewer #3 (Comments to the Authors (Required)):

A short summary of the paper, including description of the advance offered to the field.

This study nicely analyze the effect of Vdr-knockdown in K14+ cells in hair follicle and confirmed the alteration of hair cycle by the formation of cyst already at P30. The authors reported an alteration of epithelial strand length at P20 corresponding to telogen phase as Vdr-cKO mice displayed higher length in epithelial strand and consequently higher distance between HFSC and DP. Apoptosis was also reduced in the epithelial strand of Vdr-cKO mice during catagen P18 indicating that lack of Vitamin D receptor prevents the normal regression of the epithelial strand. However, the HFSC remains active as hair plucking allowed entry into anagen phase associated with an increase in the expression of Wnt and FGF genes involved in communication between DP and HFSC for anagen activation.

This study provide advance to the field as demonstrate the importance of a proper epithelial strand regression for a normal hair cycle as well as the impact of Vitamin D receptor on epithelial strand length. The study also validate that HFSC are still present but cannot be activated to push hair follicle in anagen which then induced the formation of the cyst and the alopecia.

For each main point of the paper, please indicate if the data are strongly supportive. If not, explicitly state the additional experiments essential to support the claims made and the timeframe that these would require.

- Deletion of Vdr in the epidermis disrupts hair follicle homeostasis
- what is the aim to specifically knock down Vdr in K14+ cells as compared to Vdr-null mouse?
- "Epidermal deletion of Vdr resulted in progressive alopecia, similar to that observed in Vdr-null mice (Fig. 1A)" where comes from the Vdr null mice, is it from this study or from previous study? Fig 1A compare Control versus Vdr cKO and not Vrd-null mice
- The results presented in this paragraph are not innovative as the observation made could already by find in the scientific literature e.g. Demay et al., J Steroid Biochem Mol Biol. 2008; Xie Z at al., J Invest Dermatol. 2002; Sakai Y et al., Endocrinology. 2000)
- Staining VDR/K14 to show validation of the model as compared to control mice at early time point (before cyst formation)  2 months
- If Vdr also expressed in dermal papilla as already reported, some signal should be still present in DP showing that cyst formation is induced by lack of Vdr expression in epithelial compartment

Epidermal VDR regulates hair follicle regression: data are supportive

Characteristic "surviving epithelial strands" in Vdr cKO mice are composed of Gata3+/Dst+/Krt14+ cells: the data are supportive but the part explaining the different clusters could be developed more here, how do you arrive to the conclusion that cluster 2 is a candidate for epithelial strand? Why cluster 1 and 2 were specific for vdr cKO mice? Which cluster correspond to hair follicle epithelial cell? GATA3 is also expressed in epidermis, not only IRS

Hair plucking stimulation bypasses the paused-catagen state and allows entry into anagen phase: data are supportive

VDR plays a key role in the process of hair follicle regression: Develop more about GATA3 but also Dst modulation/ expression in hair follicle. What could be the role of Dst in hair follicle (epithelial strand).

Lastly, indicate any additional issues you feel should be addressed (text changes, data presentation, statistics etc.).

Abstract:

- Vitamin D receptor (VDR) is essential for hair follicle homeostasis as its deficiency produces hair loss replace produces by induces

Introduction:

- Explain what is Cyp27b
- While Fgf5, Wnt7b, Wnt10b, and Fgf18 have been identified as factors that initiate or terminate anagen which one initiate and which one terminate
- Moreover, there have been no reported cases of impaired catagen progression in mouse models or patients with alopecia

replaced by: Moreover, no reported cases of impaired catagen progression in mouse models or patients with alopecia have been reported yet.

- demonstrates that aberrant regression that ectopically avoids cell death impairs HF regeneration demonstrates that aberrant regression that ectopically prevents cell death impairs HF regeneration

- Missing information about location of VDR expression in skin and hair follicle Mesodermal papilla cells and the outer root sheath of epidermal keratinocytes express VDR in varied degrees in correlation with the stages of the hair cycle

Results and Discussion:

Deletion of Vdr in the epidermis disrupts hair follicle homeostasis

- "However, Vdr cKO HFs did not enter anagen at P30, a stage at which they were in anagen in Vdr flox/flox (control) mice"

replace by : However, Vdr cKO HFs did not enter anagen at P30 as compared to Vdr flox/flox (control) mice.

Epidermal VDR regulates hair follicle regression

- Figure 2: Indicates which area was used to analyze Caspase+ cells

- "These findings indicated that the regression phase of Vdr cKO mice is abnormal" replace by: These findings indicated an abnormal regression phase in Vdr cKO mice...

- HFs were classified into four categories (early, mid1, mid2, and late catagen) based on the hair shaft depth (Fig. 2C).

- Figure S3: no information in legend about Figures D and E

- In addition, the Vdr cKO epithelial strand was longer than in the control after mid2 (Supplemental Fig. S4B) sentence could be improved: e.g. epithelial strand length was increased in the Vdr cKO after mid2 as compared to control whilst no significant alteration could be noticed in early and mid1 catagen

Characteristic "surviving epithelial strands" in Vdr cKO mice are composed of Gata3+/Dst+/Krt14+ cells

- What does mean LPC?

Hair plucking stimulation bypasses the paused-catagen state and allows entry into anagen phase

- Nevertheless, immunofluorescence assays revealed the presence of HF stem cells (HFSCs) identified as Sox9-positive and the bulge region as Col17-positive within the HFs, even in this paused-catagen state = sentence is unclear

Reviewer #1 (Comments to the Authors (Required)):

The authors explain that despite alopecia is commonly observed in VDDR2A patients and in mice lacking VDR, there is another mouse model lacking Cyp27b1 (and thus lacking active vitamin D) that does not display alopecia. From this fact stems the elegant hypothesis that an additional unknown function of VDR in hair follicle (HF) dynamics exists, possibly related to the modulation of catagen progression.

First, using K14 Cre Vdr cKO mice the authors narrow down the onset of HF morphological alterations to a timepoint after P15 and preceding the first postnatal anagen stage at P30. Further morphological analyses of tissue sections determined that the abnormalities follow the catagen V phase of the HF cycle. Differential length of the terminal end of the HFs in catagen motivated the study of cleaved caspase 3 expression, identifying a decrease in the number of positive cells for this marker in Vdr cKO mice coincident with retained longer epithelial strands. The scRNAseq analysis of the dorsal epidermis suggested that Vdr is involved in the elimination of the epithelial strands expressing Gata3+/Dst+/Krt14+, although this process is not completely impaired in Vdr cKO mice, in which only some cells of this type escape cell death and remain forming the surviving epithelial strands. The manuscript includes no validation of this cell population undergoing a lower level of cell death in cKO mice. Plucking-induced anagen entry occurred normally in Vdr cKO mice, indicating that the failure in hair regeneration can be bypassed.

The manuscript is well-written and structured.

Thank you very much for reviewing our manuscript and offering invaluable advice. We have addressed reviewer comments with point-by-point responses and revised the manuscript accordingly.

Major points:

- The results of this study support that the impairment in catagen progression is related to the Vdr deficiency. However, the authors suggest that this effect is independent of the interaction of the ligand (active vitamin D) with its receptor. Even though the manuscript is mostly descriptive, can the authors provide data demonstrating that the defect in HF regeneration derived from the impaired progression of the catagen phase is not reproduced by blocking the interaction of active vitamin D and its receptor in the skin of control mice?

[Response]

We appreciate the opportunity to clarify this aspect. There is no way to block the interaction of active vitamin D and its receptor in the skin. Therefore, we cannot demonstrate the reproduction of hair follicle regeneration defects in the skin of control mice. However, mice lacking Cyp27b1 an active VD synthase, and thus lacking active VD, do not display alopecia. In addition, alopecia has not been reported in patients with point mutations inside the VDR ligand pocket. These reports suggest that a ligand-independent function of VDR regulates hair follicle homeostasis. It is now mentioned in detail in the introduction of the revised manuscript (page 2, lines 39-45).

- Fig. 3 showing the "surviving epithelial strands" during the catagen stage should combine the identity markers used to define the cell subpopulation of interest at least with the cleaved caspase 3 marker. Additional approaches in this regard such as flow cytometry analysis could help to reinforce this point.

[Response]

Flow cytometry analysis presented challenges when utilizing the Gata3 antibody (CST, Cat# 5852). Nonetheless, co-staining for Gata3 and Caspase3 revealed a limited number of cells displaying dual positivity, indicating the ability of Gata3-positive cells to evade apoptosis (Fig. S5). As a response to the reviewer's comment, we have incorporated the following sentence: "However, within the Vdr cKO HF, cells expressing low levels of Gata3 exhibit resistance to apoptosis specifically in the epithelial strands" (page 5, lines 124-126). Unfortunately, performing co-staining of Dst and Caspase3 is unfeasible due to the use of antibodies derived from the same host, and there are no alternative antibodies available.

Vdr cKO (Middle catagen)

Caspase3/Gata3/DAPI

- Indicate clearly the number of animals included in the experimental groups for each experiment. This is needed to evaluate the strength of the data presented. The statistical analyses that have been used should also be clearly stated.

[Response]

We thank the reviewer for their important comments. The n number and statistical method have been included in the legend. (page 17, line 453, 454, page 18, line 477, 478, 480, 481)

Minor points:

- Indicate in the text which specific Wnt and Fgf are studied at the expression level as it is detailed in Fig. 4.

[Response]

We thank the reviewer for pointing this out and we rewritten to Wnt10a, Wnt10b (expressed in HFSCs), Fgf7 and Fgf10 (expressed in DP) (page 6, line 143-144).

Note to the Editor: I have not been able to find the supplementary figures.

Reviewer #2 (Comments to the Authors (Required)):

In this manuscript, Yudai Yoko et al., suggest the way VDR regulates the hair follicle regression. The manuscript nicely written and clearly depicts the role of VDR in regulating the hair follicle at catagen stage by showing the presence of less cell death, leaving behind the extended epithelial strand which the author calls a "surviving epithelial strand". They further elucidate with the help of scRNA analysis how VDR helps in eliminating the cells expressing Dst, Gata3 and K14 in the in the lower proximal cup region. Finally to confirm the role of VDR and the said genes in regulating the hair cycle continuation they conduct a hair plucking experiment in both control and the VDR cKO mice before the catagen stage to show indeed the process takes place due to reactivation of Fgf and Wnt under the influence of VDR.

In my opinion the manuscript is written well with subsequent results to prove the role of VDR in maintaining catagen stage hair follicle (HF). I strongly favor to accept the manuscript with said few minor changes in text or after minor clarifications.

Please change the below mentioned text or update with relevant answers:-

We thank the reviewer for these excellent comments. We have addressed reviewer comments with point-by-point responses and revised the manuscript accordingly.

- "The HFs that initially form in the dorsal skin of the mouse develop on embryonic day 12"---- I find this statement to be wrong as the mouse hair placode starts developing at E13.5-Please correct with right publication reference (Eg:- Biggs & Mikkola 2012)

[Response]

We thank the reviewer for the careful review of the manuscript. As reviewer pointed out, there was an error in the notation of embryonic day 12. The correction has been made on page 2, line 49, changing it to embryonic day 13.5, and the appropriate reference (Biggs & Mikkola, 2014) has been added.

- "These dermal cysts did not express Krt6 and Foxc1, which are expressed in the inner root sheath (IRS) of the HF (Fig. 1C, Supplemental Fig. S3A), but expressed Krt10 and Loricrin (Lor), markers of epidermal differentiation (Fig. 1D, Supplemental Fig. S3B), indicating that the dermal cysts are formed by epidermal cells" -----This context do not explain properly. Does it mean the Krt6 and Foxc1 are not expressed by epidermis? Foxc1 is known to be a HFSC marker (expressed in ORS).

[Response]

We appreciate the opportunity to clarify this point. We changed the sentence as follows to convey the meaning accurately; These dermal cysts did not express Krt6 which is known to mark the terminally differentiated companion layer and inner bulge and Foxc1 which is expressed in the HF stem cells (HFSCs), inner root sheath (IRS), isthmus, and sebaceous gland of the HF (page 3, line 74-76).

- What kind of connection is between Wnt, Fgf and Vdr? Is it directly connected to something else which is triggering the activation of Fgf and Wnt during Anagen phase?

[Response]

We appreciate the opportunity to clarify this aspect. To compensate for the lack of explanation, we added the following sentences to clarify what we mean: Moreover, no reported cases of impaired catagen progression in mouse models or patients with alopecia have been reported yet. In addition, previous reports have shown that Vdr deficiency impairs β -catenin and Lef1 mediated Wnt signaling in keratinocytes, suggesting a role for Vdr in anagen reentry (Cianferotti et al., 2007) (page 3, line 57-60).

We did not mention the relationship between Vdr and Fgf in hair follicles because there was no previous report.

- I would be happy if the author also check the expression of NFATc1 by immunostain in VDR cKO expression during different stages of hair cycle.

[Response]

We apologize to the reviewer, but we have not been able to perform immunostaining because we do not have an anti-NFATc1 antibody.

Reviewer #3 (Comments to the Authors (Required)):

A short summary of the paper, including description of the advance offered to the field.

This study nicely analyze the effect of Vdr-knockdown in K14+ cells in hair follicle and confirmed the alteration of hair cycle by the formation of cyst already at P30. The authors reported an alteration of epithelial strand length at P20 corresponding to telogen phase as Vdr-cKO mice displayed higher length in epithelial strand and consequently higher distance between HFSC and DP. Apoptosis was also reduced in the epithelial strand of Vdr-cKO mice during catagen P18 indicating that lack of Vitamin D receptor prevents the normal regression of the epithelial strand. However, the HFSC remains active as hair plucking allowed entry into anagen phase associated with an increase in the expression of Wnt and FGF genes involved in communication between DP and HFSC for anagen activation.

This study provide advance to the field as demonstrate the importance of a proper epithelial strand regression for a normal hair cycle as well as the impact of Vitamin D receptor on epithelial strand length. The study also validate that HFSC are still present but cannot be activated to push hair follicle in anagen which then induced the formation of the cyst and the alopecia.

For each main point of the paper, please indicate if the data are strongly supportive. If not, explicitly state the additional experiments essential to support the claims made and the timeframe that these would require.

Thanks to the reviewers for their summaries. We have addressed reviewer comments with point-by-point responses and revised the manuscript accordingly.

-

♣Deletion of Vdr in the epidermis disrupts hair follicle homeostasis

- what is the aim to specifically knock down Vdr in K14+ cells as compared to Vdr-null mouse?

[Response]

We appreciate the opportunity to clarify this point. Our aim is to examine the function of Vdr in the epidermis and HF separately from systemic calcium and phosphorus metabolism. In response to a comment from the reviewer, we added a similar sentence to the text. (page 3, line 69-70)

- "Epidermal deletion of Vdr resulted in progressive alopecia, similar to that observed in Vdr-null mice (Fig. 1A)" ◊ where comes from the Vdr null mice, is it from this study or from previous study? Fig 1A compare Control versus Vdr cKO and not Vrd-null mice

[Response]

We thank the reviewer for the careful review of the manuscript. As the reviewer pointed out, the data in Fig. 1A are in comparison with Vdr cKO, and the null information is from the previous study (Xie et al., 2002). We added the following sentences to clarify what we mean: Epidermal deletion of Vdr resulted in progressive alopecia similar to that observed in *Vdr* null mice by Xie et al (Fig. 1A) (Yoshizawa et al., 1997) (page 3, line 71-73).

- The results presented in this paragraph are not innovative as the observation made could already be find in the scientific literature e.g. Demay et al., J Steroid Biochem Mol Biol. 2008; Xie Z at al., J Invest Dermatol. 2002; Sakai Y et al., Endocrinology. 2000)

[Response]

We appreciate the opportunity to clarify this point. As the reviewer pointed out, alopecia due to Vdr deficiency has already been reported and is not innovative. However, since we use Vdr cKO, there is no need to consider calcium and phosphorus metabolism for analysis. On the other hand, the report by Demay et al. uses Vdr-null mice, so the analysis must take calcium and phosphorus metabolism into account.

- Staining VDR/K14 to show validation of the model as compared to control mice at early time point (before cyst formation)

[Response]

Fig. S5C shows Vdr staining data from control mice hair follicles. Also, K14 shows the results of control and Vdr cKO at the same stage in Fig. 2C. These are data before cyst formation.

- If Vdr also expressed in dermal papilla as already reported, some signal should be still present in DP showing that cyst formation is induced by lack of Vdr expression in epithelial compartment

[Response]

We thank the reviewer for this comment. The expression of Vdr in DP has been confirmed in the past (Sakai et al., 2001). But according to this paper, hair growth was reported even in VDR-null DP transplantation in keratinocyte and DP transplantation experiments. Therefore, the importance of Vdr of DP in the hair cycle is likely to be low.

♣ Epidermal VDR regulates hair follicle regression: data are supportive

[Response]

We thank the reviewer for the positive comment.

♣ Characteristic "surviving epithelial strands" in *Vdr* cKO mice are composed of *Gata3*⁺/*Dst*⁺/*Krt14*⁺ cells: the data are supportive but the part explaining the different clusters could be developed more here, how do you arrive to the conclusion that cluster 2 is a candidate for epithelial strand? Why cluster 1 and 2 were specific for *vdr* cKO mice? Which cluster correspond to hair follicle epithelial cell? GATA3 is also expressed in epidermis, not only IRS

[Response]

We thank the reviewer for this important comment. Following the reviewer's suggestion, the sentence has been changed as follows: Based on the results of the density plot, Clusters 1 and 2 were specific clusters for *Vdr* cKO mice, cluster 1 was thought to be differentiated epidermal cells due to the high expression of *Lor*, and cluster 2 was then narrowed down as the candidate of surviving epithelial strand cells. (page 5, line 114-117)

§ Hair plucking stimulation bypasses the paused-catagen state and allows entry into anagen phase: data are supportive

[Response]

We thank the reviewer for the positive comment.

§ VDR plays a key role in the process of hair follicle regression: Develop more about GATA3 but also *Dst* modulation/ expression in hair follicle. What could be the role of *Dst* in hair follicle (epithelial strand).

[Response]

We thank the reviewer for this precise comment. A previous report of scRNA-seq in anagen hair follicles revealed that *Dst* is predominantly expressed in LPC (Yang et al., 2017). However, no reports have examined the role of *Dst* in hair follicles, and its function is unknown. Following the reviewer's suggestion, the sentence has been changed as follows: *Vdr* and *Dst*, which are known to be specifically expressed in LPC of anagen (Yang et al., 2017), were also expressed in the epithelial strand of control mice (page 5, line 126-127).

Lastly, indicate any additional issues you feel should be addressed (text changes, data presentation, statistics etc.).

Abstract:

- Vitamin D receptor (VDR) is essential for hair follicle homeostasis as its deficiency produces hair loss à replace produces by induces

[Response]

We thank the reviewer's suggestion. Changed the text as suggested (page 1, line 20).

Introduction:

- Explain what is Cyp27b

[Response]

We thank the reviewers for this precise point. Following the reviewer's suggestion, the sentence has been changed as follows: Cyp27b1 an active VD synthase (page 2, line 39).

- While Fgf5, Wnt7b, Wnt10b, and Fgf18 have been identified as factors that initiate or terminate anagen ◊ which one initiate and which one terminate

[Response]

We thank the reviewers for their precise points. Following the reviewer's suggestion, the sentence has been changed as follows: Wnt7b, Wnt10b, and Fgf18 have been identified as anagen initiation factors and Fgf5 as anagen termination factor (page 3, line 55-56).

- Moreover, there have been no reported cases of impaired catagen progression in mouse models or patients with alopecia ◊ replaced by: Moreover, no reported cases of impaired catagen progression in mouse models or patients with alopecia have been reported yet.

[Response]

We thank the reviewer's suggestion. Changed the text as suggested (page 3, line 57-58).

- demonstrates that aberrant regression that ectopically avoids cell death impairs HF regeneration ◊ demonstrates that aberrant regression that ectopically prevents cell death impairs HF regeneration

[Response]

We thank the reviewer's suggestion. Changed the text as suggested (page 3, line 63-64).

- Missing information about location of VDR expression in skin and hair follicle ◊ Mesodermal papilla cells and the outer root sheath of epidermal keratinocytes express VDR in varied degrees in correlation with the stages of the hair cycle

[Response]

We thank the reviewers for this precise point. We have added the following in response to the reviewer's comments: Furthermore, *Vdr* is expressed in the outer root sheath and dermal papilla of hair follicles, and its expression is known to increase from late anagen to catagen (page 3, line 60-62)

Results and Discussion:

♣ Deletion of *Vdr* in the epidermis disrupts hair follicle homeostasis

- "However, *Vdr* cKO HF's did not enter anagen at P30, a stage at which they were in anagen in *Vdr* flox/flox (control) mice" ◊ replace by : However, *Vdr* cKO HF's did not enter anagen at P30 as compared to *Vdr* flox/flox (control) mice

[Response]

We thank the reviewer's suggestion. Changed the text as suggested (page 3, line 82-83).

♣ Epidermal VDR regulates hair follicle regression

- Figure2: Indicates which area was used to analyze Caspase+ cells

[Response]

We thank the reviewer's indication. Indicated that the experiment was performed using the back skin (page 17, line 450).

- "These findings indicated that the regression phase of *Vdr* cKO mice is abnormal" ◊ replace by: These findings indicated an abnormal regression phase in *Vdr* cKO mice...

[Response]

We thank the reviewer's suggestion. Changed the text as suggested (page 4, line 94).

- HF's were classified into four categories (early, mid1, mid2, and late catagen) based on the hair shaft depth (Fig. 2C).

[Response]

We thank the reviewer's indication. Changed the text as suggested (page 4, line 97).

- Figure S3: no information in legend about Figures D and E

[Response]

We thank the reviewer's indication. Following the reviewer's suggestion, the sentence has been changed as follows: Additionally, from around P60, *Vdr* cKO HF's lost dermal papilla (DP) (Supplemental Fig. S3C, D), gradually swelled and transformed into dermal cysts (Supplemental Fig. S3E) (page 3, line 78-80).

- In addition, the Vdr cKO epithelial strand was longer than in the control after mid2 (Supplemental Fig. S4B) \diamond sentence could be improved: e.g. epithelial strand length was increased in the Vdr cKO after mid2 as compared to control whilst no significant alteration could be noticed in early and mid1 catagen

[Response]

We thank the reviewer's suggestion. Changed the text as suggested (page 4, line 101-103).

♣ Characteristic "surviving epithelial strands" in Vdr cKO mice are composed of Gata3+/Dst+/Krt14+ cells

- What does mean LPC?

[Response]

It means the lower proximal cup that forms the lower part of the hair follicle (Sequeira et al., 2012) (page 5, line 122-123).

♣ Hair plucking stimulation bypasses the paused-catagen state and allows entry into anagen phase

- Nevertheless, immunofluorescence assays revealed the presence of HF stem cells (HFSCs) identified as Sox9-positive and the bulge region as Col17-positive within the HFs, even in this paused-catagen state = sentence is unclear

[Response]

We appreciate the opportunity to clarify this point. We added the following sentences to clarify what we mean: In other words, it is suggested that HFSCs are alive even in the paused-catagen state. (page 6, line 137-138).

August 10, 2023

RE: Life Science Alliance Manuscript #LSA-2023-02014-TR

Dr. Shun Sawatsubashi
Institute of Advanced Medical sciences, Tokushima University
3-18-15 Kuramoto-cho
Tokushima 770-8503
Japan

Dear Dr. Sawatsubashi,

Thank you for submitting your revised manuscript entitled "VDR is an essential regulator of hair follicle regression through the progression of cell death". We would be happy to publish your paper in Life Science Alliance pending final revisions necessary to meet our formatting guidelines.

- please add the Twitter handle of your host institute/organization as well as your own or/and one of the authors in our system
- please consult our manuscript preparation guidelines <https://www.life-science-alliance.org/manuscript-prep> and make sure your manuscript sections are in the correct order
- please use the [10 author names et al.] format in your references (i.e., limit the author names to the first 10)
- please add your main, supplementary figure, table, and video legends to the main manuscript text after the references section
- please remove figures from the manuscript text. Figures should be uploaded only separately, and their legends should be provided in the manuscript file only
- in our submission system, the Author Contributions selected for Masahiro Abe and Toshio Matsumoto do not qualify for authorship according to ICMJE guidelines. However, in the Author Contributions section of the manuscript, more activities have been attributed to those authors. Please either update the contributions in our system appropriately, or let us know if authors need to be removed from the manuscript.

A. FINAL FILES:

B. MANUSCRIPT ORGANIZATION AND FORMATTING:

Sincerely,

Reviewer #2 (Comments to the Authors (Required)):

The manuscript has once sent for the revision and has come back with all the needed details done as per my suggestions, hence I strongly recommend publishing in the said journal.

August 14, 2023

RE: Life Science Alliance Manuscript #LSA-2023-02014-TRR

Dr. Shun Sawatsubashi
Institute of Advanced Medical sciences, Tokushima University
3-18-15 Kuramoto-cho
Tokushima 770-8503
Japan

Dear Dr. Sawatsubashi,

Thank you for submitting your Research Article entitled "VDR is an essential regulator of hair follicle regression through the progression of cell death". It is a pleasure to let you know that your manuscript is now accepted for publication in Life Science Alliance. Congratulations on this interesting work.

DISTRIBUTION OF MATERIALS:

Again, congratulations on a very nice paper. I hope you found the review process to be constructive and are pleased with how the manuscript was handled editorially. We look forward to future exciting submissions from your lab.

Sincerely,
